# The predictive value of universal preschool developmental assessment in identifying children with later educational difficulties: A systematic review

David G. Cairney[1], Aun Kazmi[2], Lauren Delahunty[3,4], Louise Marryat[3,5]*, Rachael Wood[3]

**1** Royal Hospital for Sick Children, Edinburgh, United Kingdom, **2** Edinburgh Medical School, University of Edinburgh, Edinburgh, United Kingdom, **3** Salvesen Mindroom Research Centre, University of Edinburgh, Edinburgh, United Kingdom, **4** Institute of Health and Wellbeing, University of Glasgow, Glasgow, United Kingdom, **5** School of Health Sciences, University of Dundee, Dundee, United Kingdom

\* lmarryat001@dundee.ac.uk

## Abstract

### Background

Developmental delay affects substantial proportions of children. It can generally be identified in the pre-school years and can impact on children's educational outcomes, which in turn may affect outcomes across the life span. High income countries increasingly assess children for developmental delay in the early years, as part of universal child health programmes, however there is little evidence as to which measures best predict later educational outcomes. This systematic review aims to assess results from the current literature on which measures hold the best predictive value, in order to inform the developmental surveillance aspects of universal child health programmes.

### Methods

Systematic review sources: Medline (2000 –current), Embase (2000 –current), PsycInfo (2000 –current) and ERIC (2000 –current). Additional searching of birth cohort studies was undertaken and experts consulted.

Eligibility criteria: Included studies were in English from peer reviewed papers or books looking at developmental assessment of preschool children as part of universal child health surveillance programmes or birth cohort studies, with linked results of later educational success/difficulties. The study populations were limited to general populations of children aged 0–5 years in high income countries.

Study selection, data extraction and risk of bias assessment were carried out by two independent authors and any disagreement discussed. PROSPERO registration number CRD42018103111.

### Results

Thirteen studies were identified for inclusion in the review. The studies were highly heterogeneous: age of children at first assessment ranged from 1–5 years, and at follow-up from

**Data Availability Statement:** All relevant data are within the manuscript and its Supporting Information files.

**Funding:** This research was funded from a charitable donation to the Salvesen Mindroom Research Centre. The funder provided support in the form of salaries for authors [LM, RW]. The funders had no role in study design, data collection and analysis, decision to publish, or preparation of the manuscript.

**Competing interests:** The authors have declared that no competing interests exist.

4–26 years. Type of initial and follow-up assessment also varied. Results indicated that, with the exception of one study, the most highly predictive initial assessments comprised combined measures of children's developmental progress, such as a screening tool alongside teacher ratings and developmental histories. Other stand-alone measures also performed adequately, the best of these being the Ages and Stages Questionnaire (ASQ). Latency between measures, age of child at initial measurement, size of studies and quality of studies all impacted on the strength of results.

## Conclusions

This review was the first to systematically assess the predictive value of preschool developmental assessment at a population level on later educational outcomes. Results demonstrated consistent associations between relatively poor early child development and later educational difficulties. In general, specificity and Negative Predictive Value are high, suggesting that young children who perform well in developmental assessment are unlikely to go on to develop educational difficulties, however the sensitivity and Positive Predictive Values were generally low, indicating that these assessments would not meet the requirements for a screening test. For surveillance purposes, however, findings suggested that combined measures provided the best results, although these are resource intensive and thus difficult to implement in universal child health programmes. Health service providers may therefore wish to consider using stand-alone measures, which also were shown to provide adequate predictive value, such as the ASQ.

## Introduction

Educational failure in childhood is associated with a range of negative outcomes across the lifespan, including in relation to physical and mental health [1]. If developmental difficulties are identified early, however, timely intervention and additional support can be implemented with the aim of improving children's educational and lifecourse outcomes [2]. In recent years there has been an increasing move for high income countries to strengthen their child health programmes in order to aid early identification of children who are at risk of experiencing later developmental and educational difficulties. In Scotland, for example, additional Health Visitor reviews have been added for children aged 13–15 months, 27–30 months, and 4–5 years with a specific remit of identifying developmental delay using standardised tools [3]. As well as identifying children who are at particular risk of later difficulties, early population based surveillance in the early years is an important opportunity for health professionals to form relationships with families, particularly those who are vulnerable [4], and assess parental coping/stress, whether that be formally or informally, which evidence indicates has a significant impact on child development and poor physical and mental health outcomes in adulthood [5]. Routine developmental assessment using standardised measures also highlights developmental issues for families who may struggle to recognise or communicate concerns around their child's development. Whilst on a wider level, this population based data gives us important information about public health and future planning needs.

Although this has proven feasible to implement at a national level, several questions remain. These include what the best measure (or measures) are to use, and what the most efficient schedule of iterative assessments is, in order to ensure the highest levels of sensitivity and

specificity in identifying true difficulties, rather than simply detecting differences around levels of maturity, or transient difficulties. An overview of the literature at a population level in this area is lacking. A previous review of screening for developmental disability focused on screening within clinical settings, with relatively short-term follow-up focussing on stability of difficulties. This review, conducted 20 years ago now, found that most screens tended to over-identify children with developmental difficulties, with sensitivity rates between 45% and 72%, and specificity rates between 79% and 99% [6]. More recently, Sim et al., conducted a systematic review of the predictive validity specifically of preschool screening tools for language and/or behavioural difficulties, and indicated that language screening tools offered higher predictive validity compared with either behaviour screening tools alone, or combined language and behaviour screening tools [7]. Both of these reviews were conducted using relatively short-term follow-up periods, and had other limitations such as being implemented in clinical settings or exploring impact on school readiness, rather than longer-term educational outcomes.

## Measuring validity of screening tools

In order to be implemented as a screening tool, measurements are required to meet certain criteria, originally set out by Wilson and Junger in 1968. These criteria include that the condition should be an important health problem, that there should be an accepted treatment or intervention, that there should be a recognised early symptomatic stage, and that the natural history of the condition is known. Importantly for this study, it is also required that a suitable and acceptable test for the condition is available, and that a cut-off to implement intervention is agreed [8]. A recent review of universal health screening provision by Wilson and colleagues suggested that there is a huge amount of diversity between European countries in their implementation of screening for, or surveillance of, developmental difficulties, and that this ran parallel to a lack of evidence around the performance of these various tools [9].

Performance of assessment tools is usually measured through examination of the technical performance or concurrent validity against a gold standard. Arguably, the most important, but more uncommon assessment of a tools performance, however, is its predictive validity. This includes looking at sensitivity (of those with a condition, how many are correctly identified); specificity (of those without the condition, how many were correctly identified); positive predictive value (of those identified as having the condition, how many were correctly identified); and negative predictive value (of those not identified as having the condition, how many were correctly identified) [10]. In the current study, we also present Diagnostic Odds Ratios (DOR), where available, to aid interpretation of results. DOR of a test is the ratio of the odds of positivity in subjects with disease relative to the odds in subjects without disease. It depends significantly on the sensitivity and specificity of the test and does not depend on disease prevalence. DOR demonstrate the strength of association between the exposure and the 'disease' or condition. The DOR has particular advantages in meta-analyses as it gets around the usual issues of threshold differences due to the heterogeneity of the various measures usually being examined [11].

## Objectives

This systematic review aims to further inform the evidence base underpinning universal child health programmes through reviewing the current literature around the predictive validity of structured developmental assessments at a population level, conducted as part of universal child health surveillance programmes or birth cohort studies in the pre-school years, with linked results of later educational success and/or difficulties.

## Methods

Before beginning this study, we confirmed that no recent systematic reviews existed on this topic (S1 Fig): initially, 331 systematic reviews were assessed at title and abstract level. Following this 22 systematic reviews progressed to full text review, however, none of them addressed the question of interest.

### Protocol and registration

Our systematic review protocol was registered in advance with the International Prospective Register of Systematic Reviews (PROSPERO) on 18th July 2018 (registration number CRD42018103111).

### Eligibility criteria

Included studies were in English from peer reviewed papers or books, and looked at developmental assessment of preschool children as part of universal child health surveillance programmes or birth cohort studies, with linked results of later educational success/difficulties. The study populations were limited to general population of children aged 0–60 months in high income countries (HIC). HIC were defined using the current World Bank list of analytical income classification of economies for the fiscal year [12]. Only HIC were included in order to ensure results were applicable to strengthening and building developmental surveillance programmes within HIC. It was felt results might not be transferable between HIC and Low and Middle Income Countries.

Full inclusion and exclusion criteria are presented in Table 1.

### Information sources

Studies were initially identified by searching electronic databases Medline (2000 –current), Embase (2000 –current), PsycInfo (2000 –current) and ERIC (2000 –current). Additional studies were included by hand searching of publication lists of relevant British population-

**Table 1. Full inclusion and exclusion criteria.**

| Characteristic | Included | Excluded |
|---|---|---|
| Population | Studies involving the general population of children aged 0–60 months in high income countries | Studies focusing on selected groups of children at high risk of developmental problems<br>Studies set in low and middle income countries |
| Intervention/initial assessment | Universal preschool developmental assessment as part of broader child health programme or as part of population based birth cohort study<br>The developmental assessment may comprise any combination of eliciting parental concerns about child development, taking a developmental history, structured observation of a child's developmental level, or completing a brief developmental assessment questionnaire or assessment task | Detailed developmental assessment of children suspected of having a developmental delay or disorder<br>Studies assessing specific measures of cognitive functioning, e.g. working memory |
| Comparator | Children identified through developmental assessment as having suspected delay affecting one or more developmental domain compared to those with no suspected delay | Studies of children with sensory impairments only (hearing and/or visual impairment)<br>Studies with no control group (eg follow up studies of children with confirmed developmental delay) |
| Outcomes | Any outcome relating to educational success/difficulties such as recognition of additional educational needs, specific difficulties in literacy or mathematics, intellectual disability, or educational attainment | Diagnosis of specific developmental disorders such as ASD or ADHD |
| Study design | Cohort studies<br>These may be reported in peer reviewed published papers or books/book chapters | Narrative reviews<br>Intervention studies such as RCTs<br>Other observational designs eg cross sectional studies<br>Reports containing no new data such as editorials |

based cohort studies, specifically the 1946 National Survey of Health and Development Cohort (NSHD), the 1958 Child Development Study (NCDS), the 1970 British Cohort Study (BCS70), the Avon Longitudinal Study of Parents and Children (ALSPAC), the Millennium Cohort Study (MCS), and Growing Up in Scotland (GUS) study.

A specific search was carried out for validation studies of the Ages and Stages Questionnaire that meet the review inclusion criteria as this developmental assessment tool is used in the child health programme in the UK. There was discussion with subject matter experts (see Acknowledgements for review advisory group members) to identify missed studies. Finally, hand searching of reference lists contained within included studies was performed, in order to find other relevant studies.

## Search

We developed the search terms based on 4 main areas: 1) child development, 2) developmental assessment/screening, 3) types of developmental delay, and 4) educational attainment and learning needs/difficulties. We explored index and exploded terms which included relevant areas of child development/educational attainment (see S1 Fig for full search strategy).

## Study selection

Search results were screened to identify studies meeting inclusion criteria as specified above. Screening took place in two passes–titles and abstracts, and full text. At title and abstract screening two reviewers (LD & RW) independently assessed all papers and in cases of disagreement a resolution was sought between these two reviewers (LD & RW). At full text screening and risk of bias scoring, two reviewers (AK & DC) independently assessed all papers. Where there was disagreement a final decision was made by a third supervising author (RW). Any disagreement at the data extraction stage was discussed between the two reviewers (AK & DC) and resolution sought from a third supervising author (RW). At all points, resolution involved reference to the review protocol to ensure consistency of decision making. Agreement between the two reviewers at the title and abstract level was fair (Cohen's Kappa 0.31) [13]. At the full text review level agreement was moderate to almost perfect (Cohen's Kappa 0.55–1) depending which reviewers were compared, as three reviewers were involved at this level. Agreement was substantial for the risk of bias assessment (Cohen's Kappa 0.62).

## Risk of bias in individual studies

Study quality was assessed at the study level using a bespoke checklist (see S2 Fig) with a maximum score of 8 and categorised as high (6–8), moderate (4–5) and low (0–3) quality. The checklist was based on those provided by the Scottish Intercollegiate Guidelines Network [14] and the Critical Appraisal Skills Programme [15].

## Data extraction

Data were extracted on study identifiers, study design, study population characteristics, method and results of developmental assessment, and the methods and results of assessing educational outcomes using a bespoke data extraction template (see S3 Fig).

Study selection, risk of bias assessment and data extraction were carried out by two independent authors and discrepancies were resolved by discussion with a third supervising author.

## Summary measures

Studies were grouped into those that dichotomised the results of developmental assessment and educational outcomes, and those that treated the results of initial assessment and outcomes as continuous variables. For the first group of studies, traditional measures of test performance (sensitivity, specificity, PPV, and NPV) were extracted from the study paper, or calculated from data provided. A result of 80% or over was deemed to be a 'fair' level of specificity or specificity, with results over 90% being 'good' [16]. A Diagnostic Odds Ratio was also extracted or calculated as an overall summary measure of test performance.

The measure of association between initial developmental assessment and later educational outcomes available for the second group was more variable. Studies reported a range of outcomes including unstandardised and standardised correlation coefficients. These were extracted directly from the papers as available. No calculation of alternative measures of association was possible.

## Synthesis of results

There were insufficient comparable data available to support quantitative synthesis/meta-analysis. Due to the heterogeneous nature of the studies, formal subgroup analyses were not feasible by country of study, or method of initial developmental assessment (parental questionnaire, direct testing). For studies providing a Diagnostic Odds Ratio, this, alongside sensitivity and specificity, was examined by initial developmental domain assessed, by age at initial assessment, latency (time gap) between initial and outcome assessment, and by study quality score and size.

# Results

The database search yielded 1889 studies after removal of 156 duplicates. 339 studies were identified through reference list hand searching/expert group recommendation and 644 studies were found via cohort study hand searching. The additional Ages and Stages tool search only yielded one study (Charkaluk 2017) and this had already been identified in the database search. After title and abstract screening 47 studies underwent full text review. 34 studies were excluded, as detailed in Fig 1. The characteristics of 13 studies included are detailed in Table 2.

Of the 13 included studies, eight utilised data from population based birth cohort studies [17–24], 4 studies were population based cohort studies designed to study a developmental assessment tool [25–28] and 1 study recruited participants from a developmental cohort study [29]. It was of note that no studies were identified which were based on developmental assessments conducted as part of established child health programmes.

There was significant heterogeneity in the approach of the included studies in assessing the relationship between initial developmental assessment and later educational difficulties. The age of initial assessment ranged from 16 to 60 months and the age of educational outcome assessment ranged from 4 to 26 years. The latency between assessment and outcome was similarly varied. The studies could be broadly categorised as those with extractable dichotomous data/reported odds ratios [18,21–24,26–27] and studies with other association measures [19–20,28,30]. Three studies provided useful data in both categories [21,25,29].

## Risk of bias assessment

Sources of bias are presented in Fig 2. Inconsistent and inadequate reporting of data (i.e. different studies reporting different types of measurement) was a frequent finding across studies and limited full assessment. If there was insufficient data presented to allow assessment it was

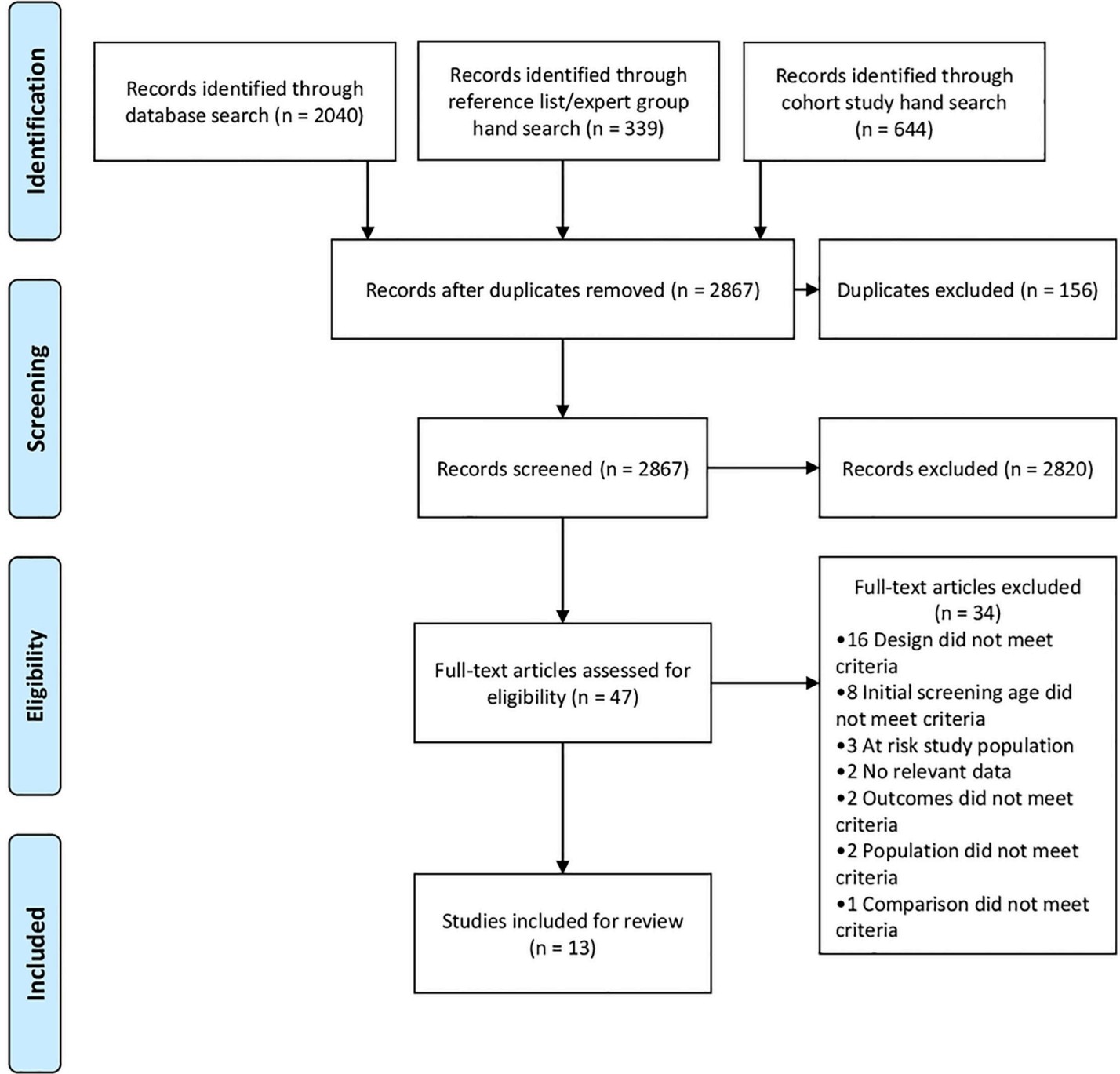

**Fig 1. PRISMA study flow chart.**

judged as a high risk of bias. Inherent in cohort studies, study retention was a significant potential source of bias across studies. The distribution of confounders was unclear in many studies although some studies accounted for this in multivariate analysis. Reporting of precision was variable however if raw values were extracted then these could be calculated independently.

Table 2. Characteristics of included studies.

| Study and publication [reference no.] | - Year of publication - Method - Setting | Study quality | Age at initial screen in months | Initial developmental assessment | | | | Educational outcome assessment | | | |
|---|---|---|---|---|---|---|---|---|---|---|---|
| | | | | Developmental domains assessed | Developmental assessment methods | Definition of developmental concern (dichotomous or continuous) | Number with initial assessment | Age at follow-up assessment in years (year) | Educational outcome type | Definition of educational difficulty (dichotomous or continuous) | Number with outcome data |
| **2.1 Studies with extractable diagnostic odds ratios** | | | | | | | | | | | |
| Bleses et al Applied Psycholinguistics [25] | 2016 Cohort study —CDI norming study Denmark | High (6) | 16–30 months | Speech, language and communication | The Danish Communicative Development Inventories (CDI) | CDI Vocabulary falling within the lowest 10% for age (dichotomous) | 3714 | 12 years | Educational attainment–subject specific | Lowest 10% for language comprehension, language decoding or reading comprehension (dichotomous) | 2863 |
| Cadman et al Canadian Medical Association Journal [26] | 1988 Cohort study —Prospective analysis of community preschool assessment Canada | High (6) | 48–60 months | Cognitive and problem solving; Speech, language and communication; Motor; Social and emotional | Denver Developmental Screening Test (DDST), kindergarten teacher's rating and health/development history. | Model 2—non-normal DDST result; Kindergarten behaviour problem; DDST language score below median; DDST fine motor/ adaptive below median Model 4— above plus Kindergarten teacher's rating of learning and behaviour problems (dichotomous) | 2761 | 8 years | Educational attainment —general | Composite measure of school performance at the end of the third school year—Child was still in grade 1 because of academic problems, was in a special education class because of school difficulties, was in grade 2 but the teacher rated the child as having a significant learning problem, or scored in the lowest 10th percentile on the Gates-MacGinitie reading test (dichotomous) | 2761 |

(*Continued*)

**Table 2.** (Continued)

| Study and publication [reference no.] | - Year of publication - Method - Setting | Study quality | Initial developmental assessment | | | | | Educational outcome assessment | | | Number with outcome data |
|---|---|---|---|---|---|---|---|---|---|---|---|
| | | | Age at initial screen in months | Developmental domains assessed | Developmental assessment methods | Definition of developmental concern (dichotomous or continuous) | Number with initial assessment | Age at follow-up assessment in years (year) | Educational outcome type | Definition of educational difficulty (dichotomous or continuous) | |
| Charkaluk et al *Pediatrics* [17] | 2017 Cohort study —Assessment of the ASQ as part of EDEN population based cohort France | High (7) | 37.3 ± 0.8 months | Communication, gross motor, fine motor, problem solving, personal-social | Parental completion of Ages and Stages Questionnaire (ASQ) as part of cohort study follow-up. | 36 month ASQ score <270 (cut off based on ROC analysis in this study) (dichotomous) | 1225 | 5 years | IQ | IQ < 85 on French version of the Weschler Preschool and Primary Scale of Intelligence– Third Edition (WPPSI-III) (dichotomous) | 939 |
| Murray et al *Annals of Neurology* [20] | 2007 Cohort study —Follow up of children in population based birth cohort study UK | High (6) | 24 months | Speech, language and communication; motor; speech, language and communication; teething included as control measure | Age of saying words other than names of parents, age of standing unaided, age of walking unaided (and age of teething) | Continuous measures | 5362 | 8 and 26 years | IQ Educational attainment —general | Nonverbal reasoning, reading comprehension (sentence completion), word pronunciation, and vocabulary (continuous) Ordinary secondary level qualifications vs advanced secondary level qualifications (dichotomous) | 3969 |

*(Continued)*

**Table 2.** (Continued)

| Study and publication [reference no.] | - Year of publication - Method - Setting | Study quality | Initial developmental assessment | | | | | Educational outcome assessment | | | | Number with outcome data |
|---|---|---|---|---|---|---|---|---|---|---|---|---|
| | | | Age at initial screen in months | Developmental domains assessed | Developmental assessment methods | Definition of developmental concern (dichotomous or continuous) | Number with initial assessment | Age at follow-up assessment in years (year) | Educational outcome type | Definition of educational difficulty (dichotomous or continuous) | | |
| Paget et al Child: Care, Health and Development [21] | 2018 Cohort study —Follow up of children in population based birth cohort study UK | Moderate (5) | 18 months 30 months 38 months 42 months | Cognitive and problem solving; speech, language and communication; motor; social and emotional Social and emotional Speech, language and communication Social and emotional | ALSPAC developed scale including items from the DDST and questions completed by the mother were used to calculate a total score including the four developmental domains Mothers were asked if they were worried about their child's behaviour development Modified version of the MacArthur Communication Questionnaire Strengths & Difficulties Questionnaire (SDQ) to assess mental health problems | Lowest 10% of total development score (dichotomous) Mother identified behavioural concern (Yes/No) (dichotomous) Lowest 10% to reflect children with poor language development (dichotomous) Dichotomized total difficulties score at the recommended clinical cut-off ($\geq 14$) (dichotomous) | 15458 | 8 and 16 years | Attendance or exclusion | Exclusion by 8 years: mothers were asked if their child had ever been excluded from school (dichotomous) Exclusion by 16 years: mothers and children were asked separately about fixed-term and permanent exclusions from school in the past 12 months. If the mother and/or child had reported any school exclusion, this was coded as "Yes" or if both the mother and child had reported no exclusions, this was coded as "No" (dichotomous) | | 8245 (8 years) 4482 (16 years) |

(*Continued*)

**Table 2.** (Continued)

| Study and publication [reference no.] | - Year of publication - Method - Setting | Study quality | Age at initial screen in months | Developmental domains assessed | Developmental assessment methods | Definition of developmental concern (dichotomous or continuous) | Number with initial assessment | Age at follow-up assessment in years (year) | Educational outcome type | Definition of educational difficulty (dichotomous or continuous) | Number with outcome data |
|---|---|---|---|---|---|---|---|---|---|---|---|
| | | | | **Initial developmental assessment** | | | | **Educational outcome assessment** | | | |
| Silva *New Zealand Medical Journal* [22] | 1981 Cohort study —Follow up of children in population based birth cohort study New Zealand | Moderate (4) | 36 months | Motor; speech, language and communication | Two items were selected from a pool of 196 items from five tests (Vineland social maturity scale, Reynell development language scale, a list of developmental milestones, a fine motor co-ordination test and the Bayley motor scales) | Unable to walk up stairs without help or without the need for watchful protection (motor problem) and/ or unable to talk in sentences of four or more syllables (language problem) | 1037 | 5 years | Composite measure | One or more of: IQ <77 on Stanford-Binet intelligence scale, score <7 on leg co-ordination subtest of McCarthy scales of children, score <40 on verbal comprehension scale of the Reynell developmental language scales, score <38 on verbal expression scale of the Reynell developmental language scales (dichotomous) | 937 |
| Smithers et al *Maternal and Child Health Journal* [23] | 2014 Cohort study —Follow up of children in population based birth cohort study Australia | High (6) | 48–60 months | Speech, language and communication; motor; social and emotional | Parent reported concerns about the child's expressive/ receptive language, or that the child had infections or hearing problems Problems with gross or fine motor skills were identified by parent report of difficulties Parents reported difficulties: (1) sleep (2) emotional wellbeing (3) energy or activity levels (4) being nervous or clingy | Parents reported hearing concern (dichotomous) Parents reported motor concern (dichotomous) One or more parent reported behavioural concerns (dichotomous) | 4386 | 6–7 years | Educational attainment– subject specific | <1 SD below the mean on the teacher-reported Mathematical Thinking subscale of the Academic Rating Scale (ARS) (dichotomous) <1 SD below the mean on the teacher-reported Language and Literacy subscales of the ARS (dichotomous) | 2883 |

*(Continued)*

**Table 2.** (Continued)

| Study and publication [reference no.] | - Year of publication - Method - Setting | Study quality | Age at initial screen in months | Initial developmental assessment | | | | Educational outcome assessment | | | |
|---|---|---|---|---|---|---|---|---|---|---|---|
| | | | | Developmental domains assessed | Developmental assessment methods | Definition of developmental concern (dichotomous or continuous) | Number with initial assessment | Age at follow-up assessment in years (year) | Educational outcome type | Definition of educational difficulty (dichotomous or continuous) | Number with outcome data |
| Valtonen et al Developmental Medicine and Child Neurology [28] | 2009 Cohort study —Prospective assessment of brief developmental assessment Finland | Moderate (4) | 48 months | Cognitive and problem solving; speech, language and communication; motor; social and emotional | Lene4 test— neurodevelopmental screening method for toddlers and preschoolers | Continuous measure | 394 | 7 years | Educational attainment —general Educational attainment —general | JLD Teacher Questionnaire— assessment of reading, writing and mathematics (continuous and dichotomous) JLD Teacher Questionnaire— assessment of language and memory skills (continuous) | 283 |
| Washbrook et al British Journal of Psychiatry [24] | 2013 Cohort study —Follow up of children in population based birth cohort study UK | High (8) | 47 months | Social and emotional | Hyperactivity/ inattention and conduct problems were assessed using the parent-completed SDQ | Subscales recoded to normal, borderline and abnormal but comparison made between normal and abnormal scores (dichotomous and continuous) | Retrospective selection, 13988 eligible at 1 year of age | 16 years | Educational attainment —general | Results from externally marked GCSE examinations: Continuous scores reflecting total points summed over the eight best GCSE grades achieved A dichotomous outcome reflecting if the pupil obtained at least five A*–C grade GCSEs including English & Maths | 11640 |

**2.2 Studies with other association measures**

*(Continued)*

**Table 2.** (Continued)

| Study and publication [reference no.] | - Year of publication - Method - Setting | Study quality | Age at initial screen in months | Initial developmental assessment | | | | Educational outcome assessment | | | |
|---|---|---|---|---|---|---|---|---|---|---|---|
| | | | | Developmental domains assessed | Developmental assessment methods | Definition of developmental concern (dichotomous or continuous) | Number with initial assessment | Age at follow-up assessment in years (year) | Educational outcome type | Definition of educational difficulty (dichotomous or continuous) | Number with outcome data |
| Duff et al Journal of Child Psychology and Psychiatry [29] | 2015 Cohort study —Follow-up of children that had taken part in BabyLab UK | Low (3) | 16–24 months | Speech, language and communication | Oxford Communicative Development Inventory completed as part of BabyLab project. | Continuous measure | 321 (939 approached) | 4–9 years | School age cognitive measures | Word reading accuracy— EOWPVT, phonological awareness— Elision subtest of CTOPP, reading accuracy— DTWRP and reading comprehension —YARC (continuous) | 300 |
| Egerton et al Centre for Longitudinal Studies (Institute of Education) [18] | 2002 Cohort study —Follow up of children in population based birth cohort study UK | Moderate (4) | 22 and 42 months | Cognitive and problem solving; speech, language and communication; motor; social and emotional | BCS70 22/42 month subgroup assessment carried out with health visitors/ completed by parents— communicative, verbal, visual-motor skills, general development, counting, comprehension, vocabulary, spatial and visual-motor skills | Continuous measures | Not stated | 10 years | Educational attainment– subject specific | Reading— Edinburgh Reading Test and numeracy— Friendly Mathematics Test (continuous) | Not stated |
| Feinstein Economica [19] | 2003 Cohort study —Follow-up of children in population based birth cohort study UK | High (6) | 22 and 42 months | Cognitive and problem solving; motor; speech, language and communication | BCS70 22/42 month subgroup assessment carried out with health visitors/ completed by parents—cube stacking, language use, personal development, drawing, counting, speaking and design copying | Continuous measures | 2457 | 10 years | Educational attainment– subject specific | Reading— Edinburgh Reading Test and numeracy— Friendly Mathematics Test (continuous) | 1292 |

(Continued)

**Table 2.** (Continued)

| Study and publication [reference no.] | - Year of publication - Method - Setting | Study quality | Age at initial screen in months | Initial developmental assessment | | | Number with initial assessment | Educational outcome assessment | | | Number with outcome data |
| --- | --- | --- | --- | --- | --- | --- | --- | --- | --- | --- | --- |
| | | | | Developmental domains assessed | Developmental assessment methods | Definition of developmental concern (dichotomous or continuous) | | Age at follow-up assessment in years (year) | Educational outcome type | Definition of educational difficulty (dichotomous or continuous) | |
| Fowler et al Developmental and Behavioural Pediatrics [27] | 1986 Cohort study —Prospective assessment of school readiness screening program USA | High (6) | 48–54 months | Cognitive and problem solving; motor | Sprigle School Readiness Screening Test (SSRST) | Continuous measure | 210 | 6 years | Reading percentile Math percentile | California Achievement score 1-2nd grade (continuous) California Achievement score 1-2nd grade (continuous) | 176 |
| Murray et al [20] | As per Table 2.1 | | | | | | | | | | |
| Valtonen et al [28] | As per Table 2.1 | | | | | | | | | | |
| Washbrook et al [24] | As per Table 2.1 | | | | | | | | | | |

CDI, The Danish Communicative Development Inventories; DDST, Denver Developmental Screening Test; ASQ, Ages and Stages Questionnaire; WPPSI-III, Wechsler Preschool and Primary Scale of Intelligence; SSRST, Sprigle School Readiness Screening Test; ALSPAC, Avon Longitudinal Study of Parents and Children; SDQ, Strengths and Difficulties Questionnaire; ARS, Academic Rating Scale; SD, standard deviation; JLD, Jyväskylä Longitudinal Study of Dyslexia; GCSE, General Certificate of Secondary Education; EOWPVT, Expressive One-Word Picture Vocabulary Test; CTOPP, Comprehensive Test of Phonological Processing; DTWRP, Diagnostic Test of Word Reading Processes; YARC, York Assessment of Reading for Comprehension.

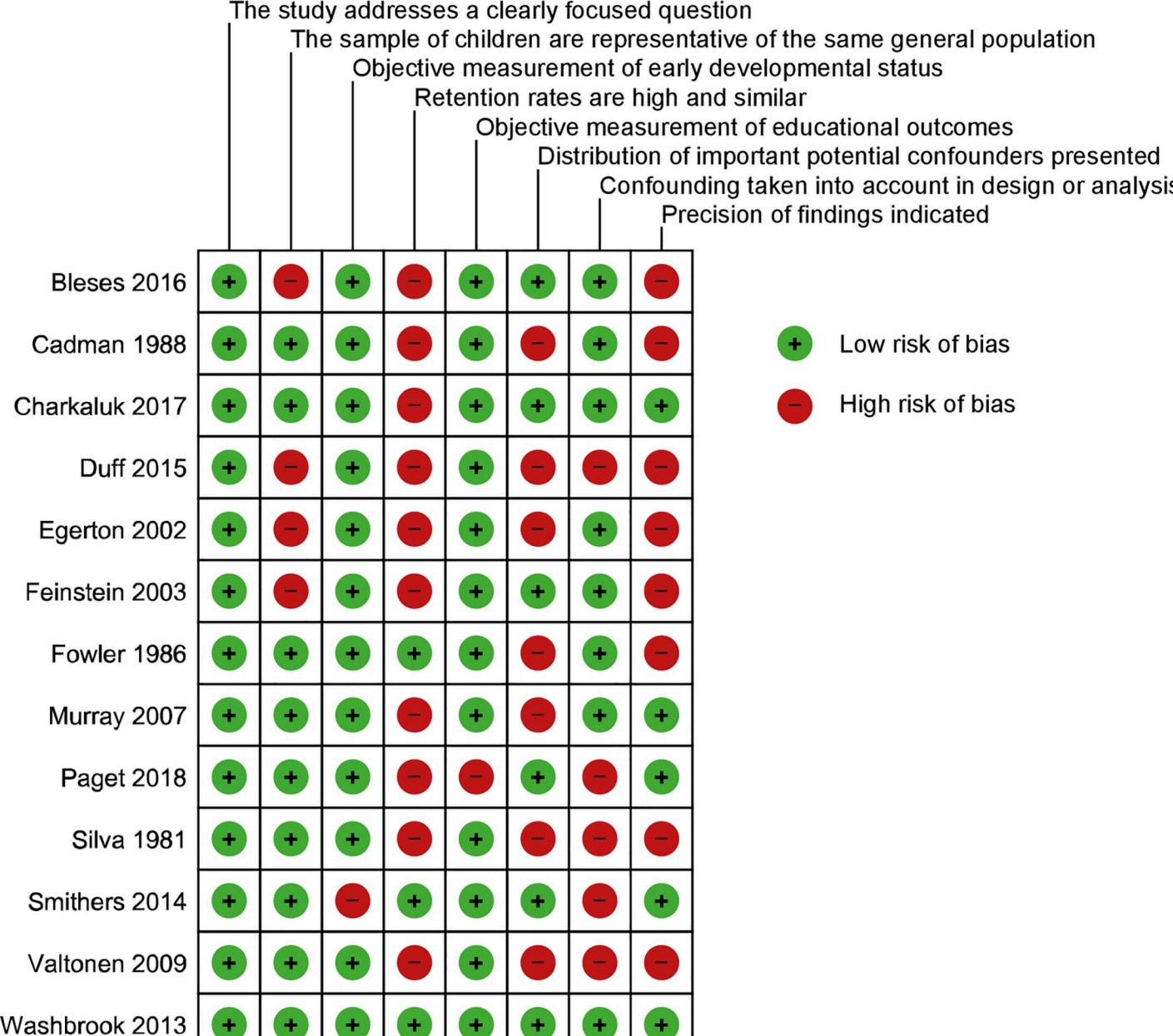

**Fig 2. Risk of bias assessment.** Risk of bias assessment using bespoke checklist based on Scottish Intercollegiate Guidelines Network and the Critical Appraisal Skills Programme checklists.

## Studies with diagnostic odds ratios

The results of the studies with calculated or quoted diagnostic odds ratios are presented in Table 3.1. Fig 3 shows diagnostic odds ratios grouped by the developmental domains assessed initially. The first group compromises studies using either general/multi-domain measures of development such as the Ages and Stages Questionnaire (ASQ) or composite measures such as issues with motor and/or speech development. Subsequent groups comprise studies initially assessing only a single developmental domain, for example children's language. Whilst almost all studies examined showed a DOR significantly above 1 (indicating a significant association

**Table 3. Study results.**

3.1 Studies with extractable diagnostic odds ratios (DOR)

| Study [Reference no.] | Initial developmental screening | | Educational outcome assessment | | DOR | 95% CI | Sens | 95% CI | Spec | 95% CI | PPV | 95% CI | NPV | 95% CI | Adjusted DOR | 95% CI |
|---|---|---|---|---|---|---|---|---|---|---|---|---|---|---|---|---|
| | Age at initial test (months) | Developmental screening methods | Age at follow-up (years) | Educational outcome methods | | | | | | | | | | | | |
| Blesses et al [25] | 16–30 months | Danish Communicative Development Inventories (CDI) lowest 10% | 12 years | Lowest 10% for language comprehension | 2.97 | 2.06 to 4.27 | 0.217 | 0.164 to 0.279 | 0.915 | 0.901 to 0.927 | 0.220 | 0.174 to 0.275 | 0.913 | 0.907 to 0.919 | | |
| | | | | Lowest 10% language decoding | 2.39 | 1.64 to 3.49 | 0.189 | 0.138 to 0.248 | 0.911 | 0.898 to 0.924 | 0.191 | 0.147 to 0.245 | 0.910 | 0.904 to 0.915 | | |
| | | | | Lowest 10% reading comprehension | 1.07 | 0.67 to 1.70 | 0.104 | 0.066 to 0.153 | 0.902 | 0.888 to 0.915 | 0.105 | 0.072 to 0.152 | 0.901 | 0.896 to 0.905 | | |
| Cadman et al [26] | 48–60 months | Model 2: Abnormal DDST; health, development and behavioural history | 8 years | Composite measure of performance at the end of the third school year | 4.45 | 3.48 to 5.70 | 0.440 | 0.389 to 0.493 | 0.850 | 0.832 to 0.867 | 0.398 | 0.350 to 0.447 | 0.871 | 0.853 to 0.887 | | |
| | | Model 4: Abnormal DDST; health, development and behavioural history; and kindergarten teacher rating | | | 10.5 | 8.15 to 13.6 | 0.590 | 0.538 to 0.640 | 0.880 | 0.863 to 0.895 | 0.525 | 0.476 to 0.574 | 0.905 | 0.889 to 0.919 | | |
| Charkaluk et al [17] | 37.3 ± 0.8 months | ASQ score < 270 | 5 years | IQ < 85 on WPPSI-III | 6.94 | 3.90 to 12.4 | 0.768 | 0.651 to 0.861 | 0.677 | 0.645 to 0.708 | 0.159 | 0.121 to 0.202 | 0.974 | 0.957 to 0.985 | 5.4[a] | 2.9 to 9.9 |
| Murray et al [20] | 24 months | Age of saying words other than names of parents | 26 years | Progression to "A" levels | | | | | | | | | | | 1.03[b] | 1.01 to 1.04 |
| Paget et al [21] | 18 months | Lowest 10% for general development | 8 years | Exclusion by 8 years | 2.01 | 0.931 to 4.34 | 0.182 | 0.082 to 0.327 | 0.901 | 0.894 to 0.907 | 0.011 | 0.005 to 0.021 | 0.995 | 0.993 to 0.996 | | |
| | | | 16 years | Exclusion by 16 years | 1.07 | 0.752 to 1.51 | 0.112 | 0.081 to 0.150 | 0.894 | 0.884 to 0.903 | 0.088 | 0.063 to 0.118 | 0.917 | 0.908 to 0.926 | | |
| | 30 months | Maternal concerns re behaviour development | 8 years | | 2.05 | 0.805 to 5.24 | 0.114 | 0.038 to 0.246 | 0.941 | 0.936 to 0.947 | 0.012 | 0.004 to 0.027 | 0.994 | 0.992 to 0.996 | | |
| | | | 16 years | | 1.90 | 1.28 to 2.82 | 0.096 | 0.066 to 0.133 | 0.947 | 0.940 to 0.954 | 0.136 | 0.094 to 0.187 | 0.923 | 0.915 to 0.932 | | |
| | 38 months | Lowest 10% for language (MacArthur Questionnaire) | 8 years | | 2.78 | 1.32 to 5.87 | 0.225 | 0.108 to 0.385 | 0.906 | 0.898 to 0.912 | 0.013 | 0.006 to 0.025 | 0.995 | 0.993 to 0.997 | | |
| | | | 16 years | | 1.62 | 1.13 to 2.32 | 0.120 | 0.087 to 0.161 | 0.922 | 0.913 to 0.931 | 0.118 | 0.085 to 0.158 | 0.924 | 0.915 to 0.932 | | |
| | 42 months | SDQ (≥14) | 8 years | | 3.86 | 2.09 to 7.15 | 0.395 | 0.250 to 0.556 | 0.855 | 0.847 to 0.863 | 0.016 | 0.010 to 0.026 | 0.996 | 0.994 to 0.997 | | |
| | | | 16 years | | 1.93 | 1.45 to 2.58 | 0.207 | 0.164 to 0.255 | 0.881 | 0.870 to 0.892 | 0.134 | 0.105 to 0.167 | 0.926 | 0.917 to 0.934 | | |
| Silva [22] | 36 months | Motor issue | 5 years | Composite measure—IQ, co-ordination, verbal skills | 5.65 | 1.81 to 17.64 | 0.052 | 0.017 to 0.116 | 0.991 | 0.981 to 0.996 | 0.385 | 0.173 to 0.652 | 0.900 | 0.896 to 0.905 | | |
| | | Language issue | | | 10.42 | 5.55 to 19.57 | 0.227 | 0.148 to 0.323 | 0.973 | 0.959 to 0.983 | 0.489 | 0.357 to 0.623 | 0.916 | 0.907 to 0.924 | | |
| | | Motor +/- language issue | | | 14.08 | 8.12 to 24.41 | 0.351 | 0.256 to 0.454 | 0.963 | 0.948 to 0.975 | 0.523 | 0.414 to 0.630 | 0.928 | 0.917 to 0.937 | | |
| | | Motor +/- language issue | | Stanford Binet IQ < 77 | 59.62 | 22.7 to 156.4 | 0.760 | 0.549 to 0.906 | 0.950 | 0.933 to 0.963 | 0.292 | 0.224 to 0.371 | 0.993 | 0.986 to 0.996 | | |
| Smithers et al [23] | 48–60 months | Parent reported language/ear/hearing problems | 6–7 years | <1 SD below the mean on maths ARS | 1.58 | 1.24 to 2.00 | 0.380 | 0.328 to 0.434 | 0.720 | 0.699 to 0.740 | 0.193 | 0.164 to 0.225 | 0.868 | 0.851 to 0.884 | | |
| | | Language issue | | <1 SD below the mean language/literacy ARS | 1.86 | 1.45 to 2.38 | 0.420 | 0.364 to 0.477 | 0.720 | 0.700 to 0.740 | 0.186 | 0.157 to 0.217 | 0.891 | 0.875 to 0.906 | | |
| | | Parent reported gross/fine motor difficulties | | Maths | 1.57 | 1.03 to 2.37 | 0.091 | 0.062 to 0.126 | 0.940 | 0.929 to 0.950 | 0.211 | 0.148 to 0.286 | 0.854 | 0.838 to 0.869 | | |
| | | | | Language/literacy | 1.96 | 1.32 to 2.94 | 0.111 | 0.078 to 0.152 | 0.940 | 0.929 to 0.950 | 0.221 | 0.158 to 0.295 | 0.874 | 0.859 to 0.888 | | |
| | | Parents reported issues with sleep, emotional wellbeing, energy/activity levels or being nervous/clingy | | Maths | 1.18 | 0.93 to 1.50 | 0.649 | 0.596 to 0.700 | 0.390 | 0.368 to 0.412 | 0.158 | 0.139 to 0.178 | 0.863 | 0.838 to 0.885 | | |

(Continued)

**Table 3.** (Continued)

| Study | Age at initial screen (months) | Developmental screening methods | Age at follow-up assessment (years) | Language/literacy | Association measure value | 95% CI | Value | 95% CI | Value | 95% CI | Value | 95% CI | Value | 95% CI | Notes/reporting concerns |
|---|---|---|---|---|---|---|---|---|---|---|---|---|---|---|---|
| Valtonen et al [28] | 48 months | Lene4 test (0.16 least stringent cut off) | 7 years | JLD Teacher Questionnaire—lowest 20% academic section | 1.04 | 0.81 to 1.34 | 0.630 | 0.573 to 0.684 | 0.380 | 0.359 to 0.402 | 0.134 | 0.117 to 0.152 | 0.871 | 0.847 to 0.892 | |
| | | | | | **5.9** | **3.1 to 11.3** | 0.696 | 0.557 to 0.808 | 0.722 | 0.659 to 0.779 | 0.380 | 0.290 to 0.480 | 0.906 | 0.865 to 0.935 | |
| | | Lene4 test (0.50 most stringent cut off) | | | **10.0** | **3.3 to 59.3** | 0.214 | 0.120 to 0.348 | 0.974 | 0.941 to 0.989 | 0.670 | 0.410 to 0.860 | 0.834 | 0.814 to 0.852 | |
| Washbrook et al [24] | 47 months | Hyperactivity/inattention problems on SDQ | 16 years | Failure to achieve five A*–C GCSEs (male) | **1.40[c]** | **1.14 to 1.72** | | | | | | | | | |
| | | | | Failure to achieve five A*–C GCSEs (female) | **1.32[c]** | **1.01 to 1.73** | | | | | | | | | |
| | | Conduct problems on SDQ | | Failure to achieve five A*–C GCSEs (male) | **1.33[c]** | **1.09 to 1.62** | | | | | | | | | |
| | | | | Failure to achieve five A*–C GCSEs (female) | 1.24[c] | 0.98 to 1.56 | | | | | | | | | |

### 3.2 Studies with other association measures

| | Initial developmental screening | | Educational outcome assessment | | | | | | | |
|---|---|---|---|---|---|---|---|---|---|---|
| Study | Age at initial screen (months) | Developmental screening methods | Age at follow-up assessment (years) | Educational outcome methods | Association measure | Association measure value | Interpretation of association | Description of variance | Adjustment for confounding/variables in model | Notes/reporting concerns |
| Duff et al [29] | 16–24 months | Oxford Communicative Development Inventory (OCDI) | 4–9 years | Word reading accuracy—EOWPVT | Standardised beta co-efficient | **0.38** | For every 1 standard deviation (SD) increase in OCDI score there is a 0.38 SD increase in word reading accuracy | OCDI score explained 16% of word reading accuracy variance | Structural equation model—included family risk and interaction between outcomes | Non-verbal IQ at school age measured but not included in model |
| | | | | Phonological awareness—Elision subtest of CTOPP | | **0.18** | 0.18 increase | 6% of variance | | |
| | | | | Reading accuracy—DTWRP | | **0.28** | 0.28 increase | 21% variance | | |
| | | | | Reading comprehension—YARC | | **0.38** | 0.38 increase | 20% variance | | |
| Egerton et al [18] | 22 months | Communicative skills | 10 years | Edinburgh Reading Test and Friendly Mathematics Test | Beta co-efficient | Reading **0.250** / Maths **0.220** | For every unit increase in communicative skills assessment there is a 0.25 unit increase in Edinburgh reading Test score and 0.22 unit increase in Friendly Mathematics Test score | Not provided | Family and schooling factors included in analysis but method of adjustment unclear | Subsamples from 2 parents families only |
| | | Verbal skills | | | | 0.092 and 0.087 | 0.092 and 0.087 increase | | | |
| | | Visual-motor skills | | | | **0.117** and **0.131** | 0.117 and 0.131 increase | | | |
| | | General development | | | | 0.177 and 0.124 | 0.177 and 0.124 increase | | | |
| | 42 months | Counting skills | | | | 0.274 and 0.246 | 0.274 and 0.246 increase | | | |
| | | Comprehension | | | | 0.255 and 0.220 | 0.255 and 0.220 increase | | | |

*(Continued)*

**Table 3.** (Continued)

| Author | Age | Domain | Outcome measure | Beta co-efficient | | Result | R² / variance explained | Adjustment | Notes |
|---|---|---|---|---|---|---|---|---|---|
| | | | | Reading | Maths | | | | |
| | | Vocabulary | | 0.163 | 0.147 | 0.163 and 0.147 increase | | Not adjusted | Subsamples from 2 parents families only |
| | | Spatial skills | | 0.147 | 0.134 | 0.147 and 0.134 increase | | | |
| | | Visual-motor skills | | 0.331 | 0.329 | 0.331 and 0.329 increase | | | |
| Feinstein [19] | 22 months | Cube stacking | Edinburgh Reading Test and Friendly Mathematics Test | 0.20 | 0.11 | For every unit increase in cube stacking score there is a 0.22 unit increase in Edinburgh reading Test score and 0.11 unit increase in Friendly Mathematics Test score | | | |
| | | Language use | | 0.22 | 0.12 | 0.22 and 0.12 increase | | | |
| | | Personal development | | 0.20 | 0.13 | 0.20 and 0.13 increase | | | |
| | | Drawing | | 0.15 | 0.14 | 0.092 and 0.14 increase | | | |
| | 42 months | Counting | | 0.29 | 0.13 | 0.29 and 0.13 increase | | | |
| | | Speaking | | 0.28 | 0.17 | 0.28 and 0.17 increase | | | |
| | | Copying designs I | | 0.32 | 0.16 | 0.32 and 0.16 increase | | | |
| | | Copying designs II | | 0.27 | 0.14 | 0.27 and 0.14 increase | | | |
| Fowler et al [27] | 48–54 months | Sprigle School Readiness Screening Test—SSRST | California Achievement score reading percentile | Not provided—only R² given | | | SSRST score explained 26% ($R^2 = 0.26$) of California Achievement reading score variance | Stepwise linear regression model. **Overall $R^2$ 0.42 (p = 0.0001).** $R^2$ for other significant factors: sex 0.02, family history 0.04, VMI 0.03, maternal education 0.07. | |
| | | | California Achievement score maths percentile | Not provided—only R² given | | | SSRST score explained 24% ($R^2 = 0.24$) of California Achievement maths score variance | Stepwise linear regression model. Overall $R^2$ 0.35 (p not reported). $R^2$ for other significant factors: VMI 0.05, maternal education 0.05, family history 0.01. | |

(Continued)

**Table 3.** (Continued)

| Study | Age | Predictor | Age at outcome | Outcome | Beta co-efficient | Description | Variance explained | Covariates adjusted | Notes |
|---|---|---|---|---|---|---|---|---|---|
| Murray et al [20] | 24 months | Age of standing unaided | 8 years | IQ | Beta co-efficient (95% confidence intervals) −0.51 (−0.71, −0.32) | Every month later a children learns to stand there would be 0.51 loss in IQ. | | Sex, socioeconomic conditions, maternal/paternal education covariates in linear and quadratic regressions | Remained significant when restricted to 'normal developers' and on quadratic regression |
| | | Age of walking unaided | | | −0.47 (−0.65, −0.30) | 0.47 loss in IQ. | | | Not significant when restricted to "normal developers" but significant on quadratic regression |
| | | Age of saying words other than names of parents | | | −0.30 (−0.40, −0.21) | 0.30 loss in IQ. | | | Remained significant when restricted to "normal developers" and on quadratic regression |
| | | Age of teething | | | −0.11 (−0.29, 0.07) | No significant association. | | | Not significant when restricted to "normal developers" or on quadratic regression |
| Valtonen et al [27] | 48 months | Lene4 test | 7 years | JLD Teacher Questionnaire—academic skills | Standardised beta co-efficient 0.65 | For every 1 SD increase in Lene4 score there is a 0.65 SD increase in JLD teacher rated academic score | Lene4 score explained 42% of JLD score | Not reported | |
| | | | | JLD Teacher Questionnaire—language skills | 0.61 | 0.61 increase | 37% of variance | | |
| Washbrook et al [24] | 47 months | Hyperactivity/inattention problems on SDQ | 16 years | Capped GCSE points (female) | Beta co-efficient (standard error) −4.04 (4.15) | Girls with an abnormal baseline hyperactivity/inattention SDQ score showed a 4 point reduction in outcome GCSE score compared to those with normal or borderline SDQ score | | Adjusted for conduct problems, IQ, parental educations, social class and early maternal depression. | 14 point reduction without adjusting for IQ |
| | | | | Capped GCSE points (male) | −10.01 (3.04) | 10 point reduction | | | 20 point reduction without adjusting for IQ |
| | | Conduct problems on SDQ | | Capped GCSE points (female) | −11.51 (4.19) | 12 point reduction | | Adjusted for hyperactivity/inattention, IQ, parental educations, social class and early maternal depression. | 18 point reduction without adjusting for IQ |

(*Continued*)

Universal preschool developmental assessment and later educational difficulties

**Table 3.** (Continued)

| | Capped GCSE points (male) | -15.23 (3.63) | 15 point reduction | 17 point reduction without adjusting for IQ |
|---|---|---|---|---|

Significant results with a p value < 0.05 are indicated in bold. No significance values were reported by Feinstein. DOR, diagnostic odds ratio; PPV, positive predictive value; NPV, negative predictive value; VMI, Beery Developmental Test of Visual Motor Integration.

[a]—Adjusted with multivariate model including sex, occupation, maternal education, etc. In children with a low ASQ score, children of mothers with a college education were significantly less likely to have IQ < 85.

[b]—Adjusted for confounders but not individually specified. Effect of speech at age 2 on secondary school attainment was no longer significant when IQ at age 8 was added into the model as an independent variable.

[c]—Adjusted for child hyperactivity/inattention or conduct problems, IQ, parental educations, social class and early maternal depression.

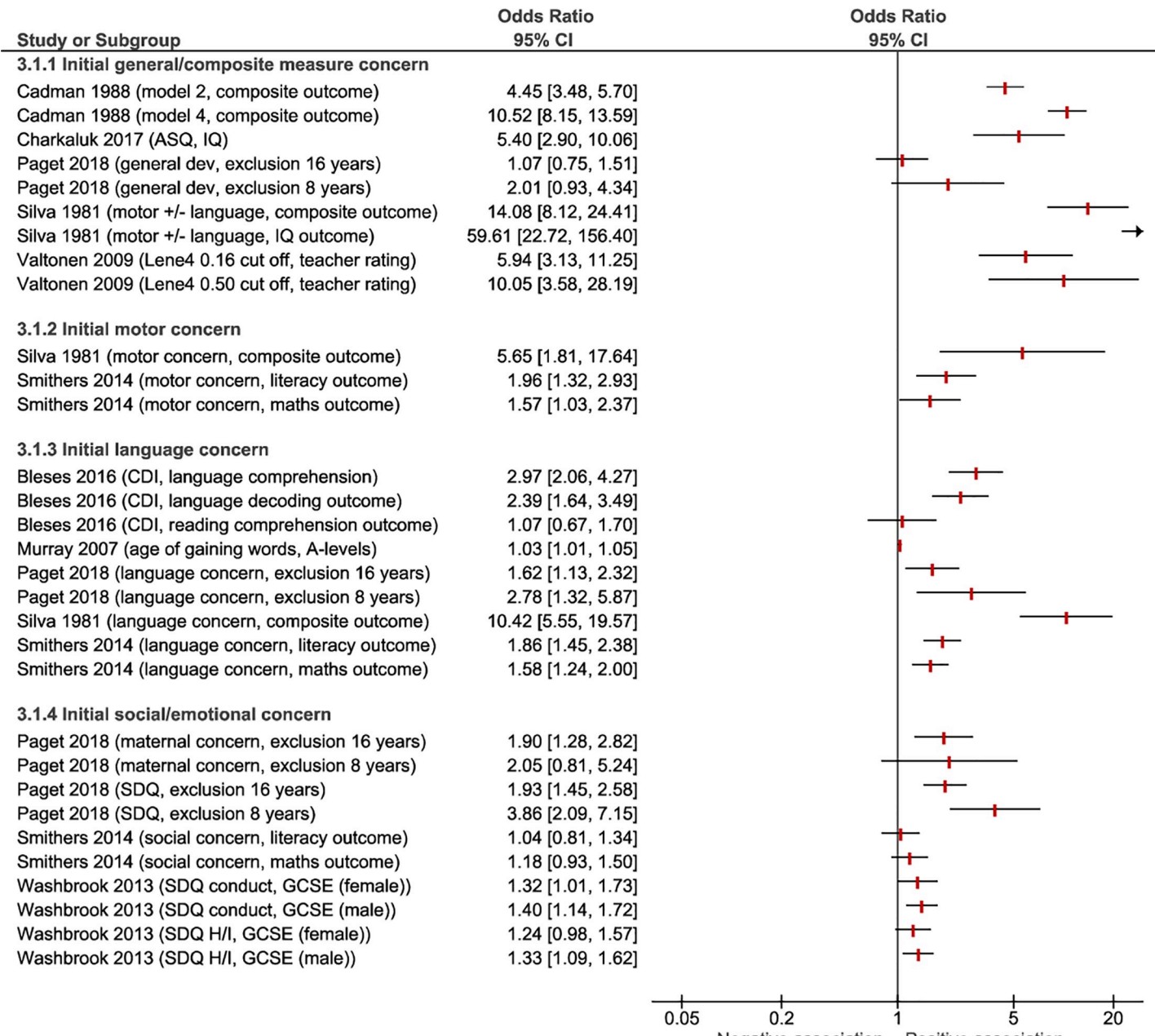

**Fig 3. Forest plot of study diagnostic odds ratios grouped by developmental domain of the initial assessment.** ASQ, Ages and Stages Questionnaire; CDI, The Danish Communicative Development Inventories; SDQ H/I, Strength and Difficulties Questionnaire Hyperactivity/Inattention score; GCSE, General Certificate of Secondary Education.

between identifying early developmental delay and later educational difficulties), studies using general/composite developmental assessment measures in general had the highest DORs. The proportion of children in each study deemed to be 'at risk' can be viewed in S2 Table.

The five highest performing measures were all found in the general/composite measures group: Silva's combined two-item assessment was the highest performing, however these data should be viewed with caution given that this assessment appears to have been selected *post hoc* based on their high predictive value within a pool of 196 items administered [22].The

combination of abnormal Denver Developmental Screening Test (DDST), a health/development/behavioural history and kindergarten teacher rating provided the next highest diagnostic odds ratio (DOR) at 10.5, with good sensitivity and specificity [25] when predicting a composite school outcome measure, but this would be resource intensive to administer in practice as part of developmental surveillance for all children. Abnormal DDST and the history component also provided a high DOR at 4.45. Interestingly DDST alone had an extremely low sensitivity at 0.06: although the authors note that the PPV was 73%, related to the high test specificity of 99% [25]. 36 month ASQ with a cut off of 270 provided a high DOR of 5.4 (adjusted) with moderate sensitivity (0.768) when predicting IQ <85 but poor positive predictive value and lower than average specificity [17]. The Lene4 test provided similar results when predicting general school academic performance [28].

Within the studies assessing children's socio-emotional development, SDQ with a cut off at ≥ 14 predicted school exclusion at 8 years in a large study with a DOR of 3.86 and sensitivity 0.395 but with an extremely low positive predictive value at 0.0163 [20]. Another study utilised the same data set from The Avon Longitudinal Study of Parents and Children (ALSPAC) cohort to demonstrate modestly significant adjusted DORs for conduct and hyperactivity problems and poor GCSE results [24]. The lowest DOR was for age of walking predicting progression to "A" levels but this adjusted value's narrow confidence intervals did not cross one suggesting a significant result.

The study with the lowest DOR was in the group of studies assessing children's motor development, with age of walking predicting progression to "A" levels (DOR 1.04, CI 0.81 to 1.34). This adjusted value's narrow confidence intervals did not cross one however, suggesting a statistically significant result [23].

Sensitivity across studies providing DORs was generally low and ranged from 0.052 [22] to 0.768 for an ASQ < 270 [28]. Over half the assessment/outcome comparisons showed specificity over 0.9 and lowest specificity was 0.390/0.380 for parent reported behaviour concerns in Smithers et al [23]. The parent reported initial measures used in this study are poorly defined and are likely to have pathologised normal variation in developmental trajectories or included insignificant hearing/middle ear problems.

## Studies with other association measures

Results of studies providing other measures of association between early developmental assessment and later educational outcomes are presented in Table 3.2.

Murray et al demonstrated that for every month later a child learns to stand they have a 0.51 loss in IQ at age 8 years [20] after adjustment for confounders. There were lesser, yet still significant, associations for age of walking and speech but not teething which was used as a control. Washbrook et al correlated conduct and hyperactivity/inattention problems on the SDQ with capped GCSE points. There was up to a 15 point penalty in GCSE scores, after adjustment for multiple confounders, including IQ; the association was strongest for males [24].

Two studies used structural equation modelling to demonstrate significant associations between developmental tests and later outcome assessments [28,29] such as the Oxford Communicative Development Inventory and later word reading accuracy.

Egerton and Bynner [18] showed significant correlations between initial developmental measures (at both 22 and 42 months) and school reading and numeracy outcomes at 10 years with adjustment for family and schooling factors. Utilising the same data set, Feinstein showed similar associations but with no adjustment and no reporting of significance values [19]. Associations for 42 month data were generally stronger in both studies but there was no clear pattern in terms of the relative performance of the developmental domains assessed.

### Age at initial assessment, test latency, study quality and study size

The relationship between the age at initial developmental assessment; the length of the latency period between the initial developmental assessment and the subsequent assessment of educational outcomes; study quality; and study size, and study findings (in terms of diagnostic odds ratio, sensitivity and specificity) was explored (Fig 4). Higher study quality and larger study size was associated with lower diagnostic odds ratios. There was no association between age at initial developmental assessment and the strength of the diagnostic odds ratio. Higher age at initial assessment was associated with higher sensitivity but lower specificity.

Overall, a shorter latency period between initial and subsequent assessment was associated with higher diagnostic odds ratios and sensitivity. There was no association between latency and specificity. To examine this further, S3 Fig. provides additional plots showing the relationship between latency and study findings for studies with age at initial developmental assessment under, and over, 36 months separately. A shorter latency period between initial and subsequent assessment was associated with higher diagnostic odds ratios in studies involving initial assessment of children aged under, and over, 36 months, however the association was only significant for children initially assessed at under 36 months.

## Discussion

This paper is the first, as far as we are aware, to systematically review studies exploring associations between early developmental assessments at a population level, and later educational outcomes, in order to better inform universal child health programmes. The review aimed to explore the psychometric properties of existing developmental surveillance tools being used in high income countries to evaluate their use in identifying developmental difficulties, and to guide future policy decisions for high income countries refining such programmes. Findings suggested a myriad of approaches which could be used within a universal child health programme to assess developmental difficulties in the preschool years. Results were not straightforward. Early developmental measures were found to be associated with later educational outcomes, however with different degrees of strength, dependent on factors such as the type of developmental measure used, the time lag between initial assessment and follow-up and the ages of the children's initial assessment. The type of initial developmental assessment measure showing the strongest association between early development and later educational outcomes was Silva's two-item assessment, however, this should be treated with caution as it was selected on a post-hoc basis from more than 150 different measures [22]. Aside from this rather unusual case, the other best performing measures tended to be fairly broad or combined measures, encompassing a variety of different domains. This is in contrast to Sim et al's findings on the predictive validity of language and/or behavioural measures in the preschool years, which suggested that language measures alone best predicted later outcomes, compared with either behavioural measures alone or combined measures [7]. This may be reflective of the different measures being examined: Sim et al. looked specifically at language and/or behaviour as predictors of later development, compared with our focus on developmental delay, which may or may not encompass language and/or behaviour among other elements. Furthermore, the outcomes examined differed, with Sim et al., looking for associations with developmental delay around the start of school, compared with the current study which explored predictors of later educational difficulties. It may be that these are fundamentally different, with factors associated with later educational delay being broader than those associated with relatively early developmental delay. Interestingly, the highest performing of these combined measures in the current study was the Denver Developmental Screening Test (DDST), in combination with other measures such as developmental histories and kindergarten teacher ratings [25]. Of

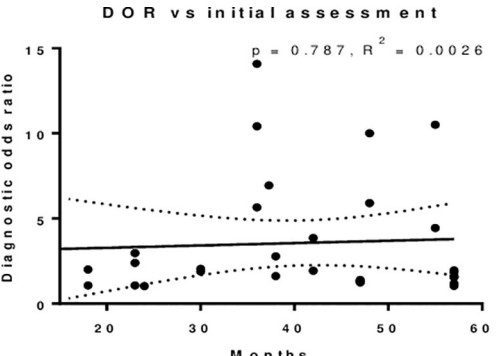

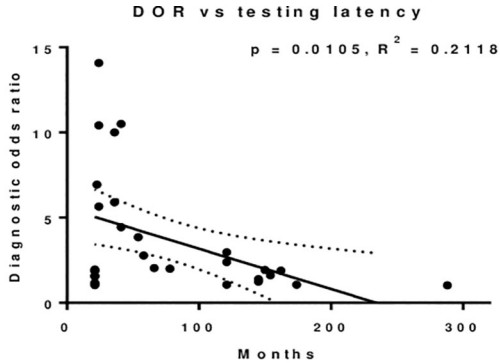

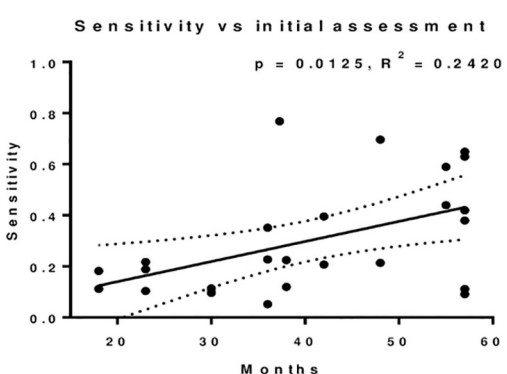

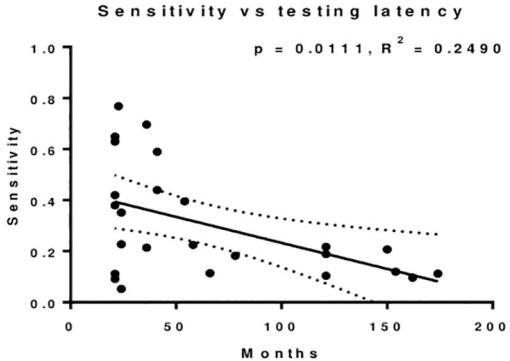

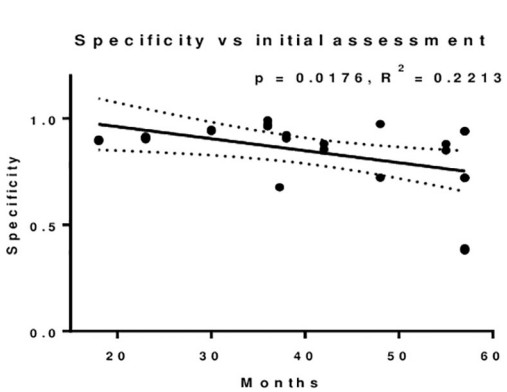

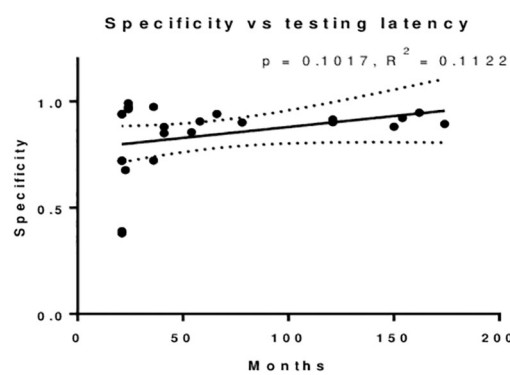

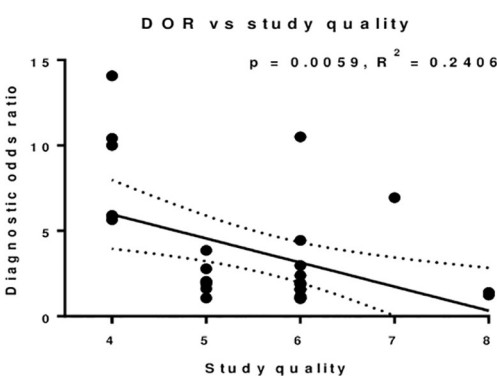

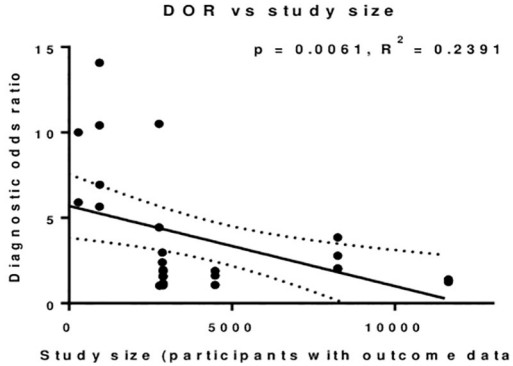

**Fig 4. Scatter plots of diagnostic odds ratios, sensitivity and specificity versus age in months at initial assessment and latency between initial assessment and school assessment in months.** Data are from the studies included in Table 3.1. Diagnostic odds ratios versus study quality and study size are also assessed. Linear regression is demonstrated with p values (results < 0.05 in bold) and $R^2$ values for goodness-of-fit indicated.

course the issue in a universal child health programme is the resource involved in reporting and interpreting measures which include resource-intensive elements such as administration of standardised tools.

Tools which are perhaps logistically easier to carry out at a population scale include the reaching of developmental milestones in infancy and early childhood. Murray, for example, demonstrated a small but significant effect of reaching developmental milestones on later IQ [20], while Charkaluk's study demonstrated that the ASQ (the current tool of choice in the UK child health programme) provided good Diagnostic Odds Ratios as well as a reasonable level of sensitivity [17]. In addition, the Conduct Problems and Hyperactivity/Inattention components of the SDQ were found to predict later GCSE success to a reasonable degree [24]. A further study found that the inattention element of hyperactivity and inattention is the most important in predicting later educational success out of the two, and thus it may be possible to reduce this screen further [31].

As may be expected, an effect of latency between initial assessment and outcome measurement was apparent, whereby shorter time periods between initial assessment and follow-up led to better predictive validity. Examination of results of initial assessment before and after 36 months indicated that the effect of latency was more prominent when the assessment was conducted at an earlier age. At this young age of initial assessment, developmental trajectories are more variable, and a single screen will only provide a snapshot into a dynamic process [2]. Related to this is the indication within our findings that later initial assessments are more reliable in detecting difficulties, likely related to them being closer to the outcome measurement in many case, although this may also relate to the increased maturity of the children being measured, and, relatedly, the increased stability in developmental trajectories: children whose skills appear normal at an early age may yet demonstrate problems later on, for example a child may have good motor, communication and social skills at age three, but may develop difficulties with reading at age six, whilst, conversely, younger children may also show apparent transient delays, before they subsequently catch-up [32]. Sim et al., found a similar effect in terms of time lapse between first and follow-up assessments. They did not, however, find an association between age of child at first assessment and performance of assessment, perhaps related to the slightly older age of initial assessment reported within the Sim et al. study, compared with the current research (2–6 year olds in Sim et. al vs. 16–60 months in the current study) [7].

It is important to note, however, the effect of potential bias within the studies being reviewed. This was apparent when examining the relationship between study size, study quality, and results of these studies. Study quality and size were both demonstrated to account for around a quarter of the variance seen between different studies, with higher quality and larger studies reporting lower Diagnostic Odds Ratios on average than lower quality and smaller studies. This latter finding is in line with results from other systematic reviews across a range of topics, which have reported trends towards lower effect sizes being associated with higher quality, and larger, studies [32,33].

## Strengths and limitations

The key strength of this study is the rigorous approach to systematically reviewing the literature on the screening of developmental difficulties in the population for later educational

difficulties. Alongside a thorough search and screening process of journal articles, the authors also explored the reference lists of included studies and consulted with key experts in the field.

In terms of limitations of the review, the resource available meant that only English-language papers were reviewed. In addition, studies were limited to high income countries, which may mean that results are not generalizable to low and middle income countries. Tools were also not examined in terms of their impact on inequalities, and may reflect bias in terms of ethnicity and socio-economic classification: this should be considered prior to any implementation. The review is, of course, limited by the quality of the studies available within the review: this included limited information required to assess study quality, as well as inconsistency in the reporting of data items, such as diagnostic odds ratios. In addition, the variability of both initial and outcome assessments make the synthesis of results difficult. Data used in the scatter plots (Fig 4 and S3 Fig) were derived from data in Table 3.1, and so included combined multiple data points for some studies, as well as heterogeneous initial and follow-up assessments.

## Conclusions

This study is the first to systematically review the evidence around the strength of association between developmental difficulties in the general pre-school population in relation to later educational outcomes. Overall, results clearly showed an association between early developmental delay and later educational difficulties. The strengths of such associations varied, depending on the detail of the initial developmental assessment method, the exact educational outcome examined, and ages of children's initial assessment and time lags between initial and follow-up assessment. In terms of the initial developmental assessment used, results indicated that Silva's two-item test demonstrated the best performance in relation to predicting later educational outcomes, however, the post-hoc nature of the selection of this screen leads us to caveat the result. Second to Silva's test, the best performing results were of a very different nature: they were primarily combinations of measures involving different components. Some of these may be practically difficult to implement as part of a universal child health programme, for example, the DDST plus developmental history and kindergarten teacher ratings, as this would require substantial investment in both time and money. Other assessment tools, such as the ASQ or SDQ Externalising Behaviours measures, which are far quicker and easier to implement, also provided adequate predictive value, suggesting that these may be a good compromise for high income countries investing in identifying children at risk of educational difficulties through a universal developmental surveillance programme. Finally, these results suggest that the age at which children are assessed may be important, with the predictive value decreasing, the younger the assessment is carried out, along with the longer the time period between initial measurement and follow-up. This may indicate the requirement for assessment to occur at various stages in the developmental pathway, rather than at just one time point, in order to identify meaningful and reliable results.

## Supporting information

**S1 Fig. Search strategy.** Medline search strategy for example. Date: 15/06/2017. (TIF)

**S2 Fig. Risk of bias assessment.** All questions are scored as yes (= low risk of bias (LROB)) or no (= high risk of bias (HROB)). Questions A is a stop/go question, i.e. if scored no/HROB, the study would not be included further in the review. Included studies are then scored against questions 1–8. Results are categorised as:

- 6–8 questions scored as yes/LROB = high quality

- 4–5 = moderate quality

- 0–3 = low quality.
  (TIF)

**S3 Fig.** Scatter plots of diagnostic odds ratios, sensitivity and specificity versus latency between initial assessment and school assessment in months: Results shown separately for studies with initial developmental assessment conducted prior to 36 months (left) and at 36 months or later (right). Linear regression is demonstrated with p values (results < 0.05 in bold) and $R^2$ values for goodness-of-fit indicated.
(TIF)

**S1 Table. Bespoke data extraction template for included studies.** Template used to extract data for studies included in the review.
(PDF)

**S2 Table. Percentage of at risk individuals on preschool assessment and incidence of adverse educational outcomes.**
(PDF)

## Acknowledgments

We would like to thank our review advisory group for their expert views on papers which should be included: Prof Anne O'Hare, University of Edinburgh, Prof Phil Wilson, University of Aberdeen, Prof James Law, University of Newcastle, Dr Lucy Thompson, University of Glasgow/University of Aberdeen, Dr Anna Pearce, University of Glasgow, and Dr Fiona Sim, University of Glasgow.

## Author Contributions

**Conceptualization:** David G. Cairney, Rachael Wood.

**Formal analysis:** David G. Cairney, Aun Kazmi, Lauren Delahunty.

**Investigation:** Aun Kazmi, Lauren Delahunty.

**Methodology:** Lauren Delahunty.

**Project administration:** David G. Cairney, Aun Kazmi.

**Supervision:** Louise Marryat, Rachael Wood.

**Validation:** Rachael Wood.

**Writing – original draft:** David G. Cairney, Louise Marryat.

**Writing – review & editing:** David G. Cairney, Aun Kazmi, Lauren Delahunty, Rachael Wood.

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
