## [Decision Letter · Decision Letter 0]

20 Oct 2020

PONE-D-20-12370

The predictive value of universal preschool developmental assessment in identifying children with later educational difficulties: a systematic review

PLOS ONE

Dear Dr. Marryat,

Thank you for submitting your manuscript to PLOS ONE. After careful consideration, we feel that it has merit but does not fully meet PLOS ONE’s publication criteria as it currently stands. Therefore, we invite you to submit a revised version of the manuscript that addresses the points raised during the review process.

I was asked to handle your manuscript after the original editor assigned to this paper became unavailable.  I have now received two expert reviews, both of which highlight the importance of the question you pose and the soundness of the methods you used to address it.  Both reviewers do have a number of additional questions that they would like to be addressed in a revision; I suspect that you and your colleagues will be able to provide satisfactory responses to these concerns without difficulty.  In general, however, I would like to commend you and your colleagues for your commitment to methodological rigor and transparency: values that are of utmost importance at PLOS One.

I will add several other comments that arose during my own reading of the manuscript.  I echo R1's recommendation about the benefit of a clearer organizational strategy in the Results, and R2's concerns about the age/timing factors (see more below).  Both reviewers also asked for several details that, in my reading, were already present in the manuscript.  In such cases, please feel free to direct the reviewer's attention to where the content was provided (although do also feel free to make this more obvious as desired).  In places where the requested content was not originally included, please do supply it. 

A reply letter that responds to each point raised by the academic editor and reviewer(s). You should upload this letter as a separate file labeled 'Response to Reviewers'.A marked-up copy of your manuscript that highlights changes made to the original version. You should upload this as a separate file labeled 'Revised Manuscript with Track Changes'.An unmarked version of your revised paper without tracked changes. You should upload this as a separate file labeled 'Manuscript'.

We look forward to receiving your revised manuscript.

Kind regards,

Matthew L. Hall, PhD

Academic Editor

PLOS ONE

Journal Requirements:

2.In your Methods section, please specify the date of your search.

3.Thank you for stating the following financial disclosure:

 [This research was funded from a charitable donation to the Salvesen Mindroom Research Centre. The funder had no role in the design, conduct, or reporting of the review.].

We note that one or more of the authors is affiliated with the funding organization, indicating the funder may have had some role in the design, data collection, analysis or preparation of your manuscript for publication; in other words, the funder played an indirect role through the participation of the co-authors. If the funding organization did not play a role in the study design, data collection and analysis, decision to publish, or preparation of the manuscript and only provided financial support in the form of authors' salaries and/or research materials, please do the following:

Review your statements relating to the author contributions, and ensure you have specifically and accurately indicated the role(s) that these authors had in your study. These amendments should be made in the online form.

Confirm in your cover letter that you agree with the following statement, and we will change the online submission form on your behalf:

Additional Editor Comments (if provided):

Major comments

The apparent contrast between the results of Sim et al. (2019) and the present study is worth further discussion. To what extent are the findings reconcilable? For example, it seems possible that predicting language and behavioral difficulties might naturally require a different screening/assessment approach than predicting educational outcomes. In addition, the screening tools that were analyzed in the present study vary in the extent to which they load on language. If public health programs aim to develop research-based screening protocols, they will likely be interested in identifying the smallest set of instruments that predict the widest range of outcomes. Considering the results of Sim et al. (2019) alongside those of the present study, what recommendations or cautions are in order?

I would like to see a more thorough discussion of the potential dissociability of age at initial screening, age at outcome measurement, and elapsed time. It is unsurprising that less elapsed time results in greater predictive validity. However, it is presently unclear whether the lower predictive validity of screening at younger ages is actually reflective of developmental phenomena, as the manuscript currently suggests, or whether this is a confound of earlier screenings being used to make predictions across longer timespans. I would encourage the authors to either address this point more directly, or to explicitly state what prevents this question from being addressed. Again, those responsible for designing screening programs will surely want to know whether to, for example, attempt to screen once early on, once later on, or twice (where the earlier screen predicts on a much shorter timespan). It seems to me that the present data may be able to speak to that question; I encourage the authors to comment on it.

I would like the authors to be more explicit about their vision for the practical application of the knowledge gained from this study. For instance, I initially wondered whether the goal was to understand what high-income countries are doing so that low-income countries might learn from their example, but this was (in my view, rightly) recognized as a limitation later on. Is the goal to make recommendations for high-income countries who are not yet doing early surveillance? To help countries that are doing population-based screening do so more efficiently? Something else?

I write from an American perspective, where there is considerable concern that many of our standardized outcome measures unjustly reflect prejudices regarding differences in race, ethnicity, and class, among others. Thus, some of the variability in predictive accuracy is attributable to measurement error (in screening and/or eventual outcomes). I invite the authors to comment on the extent to which this may be a problem in the extant research. For example, to what extent to the populations tested resemble the norming samples of the measures used? To what extent have the measures themselves been evaluated to determine whether they inappropriately penalize or pathologize diversity?

Additional comments

Introduction: The manuscript makes a strong case for the expected impact of early population-based surveillance/assessment, but no information is provided about the ultimate impact of these efforts. Including this type of evidence would help non-specialists to understand the value of the present study and its potential long-term impact.

Table 1: What was the rationale for excluding diagnostic studies, if they met the other inclusion criteria?

Information Sources: In the introduction, the authors (justifiably) critique a previous review for having been conducted 20 years ago. What then is the rationale for including studies from the mid-20th century, especially given the substantial changes to our understanding of child development, disorders, and diagnostics since that time?

Data Extraction: Data on the rates of initial agreement between the coders would be appreciated, either as supplemental material, or in summarized form in the main text.

Typographical comments:

There are a number of small syntactic or word-level errors throughout the manuscript (e.g. ln. 74-76; ln 290, inter al.). Please proofread carefully in preparation for the next submission.

Similarly, there are several instances where a study is named without an accompanying citation number.

Reviewers' comments:

Reviewer's Responses to Questions

**Comments to the Author**

1. Is the manuscript technically sound, and do the data support the conclusions?

Reviewer #1: Partly

Reviewer #2: Yes

2. Has the statistical analysis been performed appropriately and rigorously? 

Reviewer #1: Yes

Reviewer #2: Yes

3. Have the authors made all data underlying the findings in their manuscript fully available?

Reviewer #1: Yes

Reviewer #2: Yes

4. Is the manuscript presented in an intelligible fashion and written in standard English?

Reviewer #1: Yes

Reviewer #2: Yes

5. Review Comments to the Author

Reviewer #1: The manuscript is well written and presents data on a question that is important for the field of communication disorders and sciences. These findings are important as we move forward and continue to develop various child-find and early intervention screening measures. The comments and suggestions made below are meant to add to the rigor of the paper, ensure that these findings can be replicated, and that the methods can be used in similar studies. All of these comments are justifications for my decision in question one.

The authors should describe what are "acceptable" or "good" levels of sensitivity and specificity. I would suggest that the authors done by Plante and Dollaghan. .

The the method section specifically lines 118-121 you should give your inclusionary criteria for your initial search.

The authors should describe what is the criteria to be considered a "high income country"? Authors should provide a rationale for only including high income countries.

The authors discuss doing a special search just for articles that used the Ages and Stages Questionnaire. Why is this? Is there potential bias for the results for doing an additional search for one screening tool and not the others? This needs to be grounded in a theoretical framework and should be considered as a limitation is this is a potential source for bias.

A rationale and framework for why the specific search terms were selected is needed so that the rigor can be evaluated and methods and results can be replicated.

Lines 202-204: What is meany by "inconsistent and inadequate reporting"? This is important to fully evaluate the methods used and to make sure important article findings were not discarded.

Line 209: Why are the combination of screening tools "cumbersome to administer"? This needs to be more clear and should be given more attention in the discussion as this is of critical importance for implementation.

The results need to be more organized and more clear. The authors seem to jump back and forth between the screening tools and their diagnostic ability/accuracy. I would suggest given more general results and then give specific results in section for each screening tool you are discussing to make sure that this is easier to read for the audience.

Reviewer #2: This article addresses an important question, and the systematic review is well conducted, although it does not have a clear conclusion due to the heterogeneity of the studies included.

I have some comments:

- The research is about “universal preschool developmental assessment”. I understood it refers to the same framework as “screening of developmental difficulties in the population”. In this context, predictive value of baby-tests is not studied. The authors should more clearly set this point from the beginning of the article.

- About full inclusion and exclusion criteria (table 1), age range is specified for the population, ie 0-60 months. But no age range is specified for outcomes. In the text, it is quite surprising to see that there is an overlap between age range for preschool assessments and later educational difficulties, which begins at 4 years in ref [24].

- In the same field, there is a latency of only 6 to 12 months in ref [22]. Is this study really in the topic of the current paper?

- For each study except those using continuous measures, it would be useful to have the information of how many children are considered “at risk” or “below threshold” at the preschool developmental assessment. Considering the public health perspective adopted by the authors, this is an important information. The prevalence of the outcome, when dichotomous, is also necessary to interpret these data.

- Less important: definitions of sensitivity, specificity and predictive values is given, but not the meaning of DOR. Could you consider giving all definitions?

- Also less important: the beginning of the Discussion is very similar to the introduction. It is probably possible to jump more quickly to the summarize of the results (although difficult to summarize….)

6. PLOS authors have the option to publish the peer review history of their article (what does this mean?). If published, this will include your full peer review and any attached files.

Reviewer #1: No

Reviewer #2: No

---

## [Author Response · Author response to Decision Letter 0]

4 Dec 2020

Please see the cover letter and response to reviewers.

---

## [Decision Letter · Decision Letter 1]

21 Dec 2020

PONE-D-20-12370R1

The predictive value of universal preschool developmental assessment in identifying children with later educational difficulties: a systematic review

PLOS ONE

Dear Dr. Marryat,

Thank you for resubmitting your manuscript to PLOS ONE. Both reviewers and I agree that it has improved significantly; as you will see from the comments below, only several relatively minor points remain. 

I concur with R1 that the kappa values for agreement at the title and abstract level are low; however, my own experience with systematic reviews has been similar.  I would request that you provide more information about the coding procedures: for instance, were *all* titles and *all* abstracts coded by at least two people, or only a subset?  When coders disagreed at this level, what processes and/or principles were used to reach a resolution?  Being explicit about this process will be particularly important if only a subset of titles and abstracts were coded by more than a single rater.  It would also be helpful to provide information about the specific criteria that the raters were using to evaluate titles and abstracts. 

Please do also attend to the remaining comments, and submit your revised manuscript by Feb 04 2021 11:59PM. If you will need more time than this to complete your revisions, please reply to this message or contact the journal office at plosone@plos.org. Please include the following items when submitting your revised manuscript:

A reply letter that responds to each point raised by the academic editor and reviewer(s). You should upload this letter as a separate file labeled 'Response to Reviewers'.A marked-up copy of your manuscript that highlights changes made to the original version. You should upload this as a separate file labeled 'Revised Manuscript with Track Changes'.An unmarked version of your revised paper without tracked changes. You should upload this as a separate file labeled 'Manuscript'.

We look forward to receiving your revised manuscript.

Kind regards,

Matthew L. Hall, PhD

Academic Editor

PLOS ONE

Reviewers' comments:

Reviewer's Responses to Questions

**Comments to the Author**

1. If the authors have adequately addressed your comments raised in a previous round of review and you feel that this manuscript is now acceptable for publication, you may indicate that here to bypass the “Comments to the Author” section, enter your conflict of interest statement in the “Confidential to Editor” section, and submit your "Accept" recommendation.

Reviewer #1: (No Response)

Reviewer #2: All comments have been addressed

2. Is the manuscript technically sound, and do the data support the conclusions?

Reviewer #1: Yes

Reviewer #2: Yes

3. Has the statistical analysis been performed appropriately and rigorously? 

Reviewer #1: Yes

Reviewer #2: Yes

4. Have the authors made all data underlying the findings in their manuscript fully available?

Reviewer #1: Yes

Reviewer #2: Yes

5. Is the manuscript presented in an intelligible fashion and written in standard English?

Reviewer #1: Yes

Reviewer #2: Yes

6. Review Comments to the Author

Reviewer #1: Overall this manuscript has greatly improved and is highly relevant and needed work in this specific area. There are a few minor areas to consider and are outlined below. First, for the study selection the kappa levels appear a bit low and do vary quite a bit based on what was presented. The authors should address why these values vary and why the .31 is considered good, when others would consider this low or weak agreement. How does this impact the results and the study limitations? A justification for the "bespoke checklist" is needed. The authors should discuss how this impacts the study especially in terms of design and rigor. Finally there remains some issues concerning the diagnostic accuracy levels used. The authors state that they have used an 80% diagnostic accuracy rating and have labeled this as a "good" accuracy level. However, work by Plante and Vance (1994) as well as other state that 80% accuracy is fair and 90% is good. I would suggest that the authors do a more complete reading and justification of this decision in their study design.

Reviewer #2: Thank you for your additionnal work on this paper, which makes the message clearer.

however, "baby-tests" does non refer to newborn examination, but to developmental scales, developped from Gesell works. Baby-tests include Balyley scales for example.

I understood that you chose not to include them because they cannot be part of universal screening programs. Is it true? I may be worth specify.

7. PLOS authors have the option to publish the peer review history of their article (what does this mean?). If published, this will include your full peer review and any attached files.

Reviewer #1: No

Reviewer #2: No

---

## [Author Response · Author response to Decision Letter 1]

22 Jan 2021

Response to reviewers

Editor comments: I concur with R1 that the kappa values for agreement at the title and abstract level are low; however, my own experience with systematic reviews has been similar. I would request that you provide more information about the coding procedures: for instance, were *all* titles and *all* abstracts coded by at least two people, or only a subset? When coders disagreed at this level, what processes and/or principles were used to reach a resolution? Being explicit about this process will be particularly important if only a subset of titles and abstracts were coded by more than a single rater. It would also be helpful to provide information about the specific criteria that the raters were using to evaluate titles and abstracts.

We have added into the study selection section the following text providing further detail about coding procedures:

At title and abstract screening two reviewers (LD & RW) independently assessed all papers and in cases of disagreement a resolution was sought between these two reviewers (LD & RW). At full text screening and risk of bias scoring, two reviewers (AK & DC) independently assessed all papers. Where there was disagreement a final decision was made by a third supervising author (RW). Any disagreement at the data extraction stage was discussed between the two reviewers (AK & DC) and resolution sought from a third supervising author (RW). At all points, resolution involved reference to the review protocol to ensure consistency of decision making. 

Reviewer #1: Overall this manuscript has greatly improved and is highly relevant and needed work in this specific area. There are a few minor areas to consider and are outlined below. First, for the study selection the kappa levels appear a bit low and do vary quite a bit based on what was presented. The authors should address why these values vary and why the .31 is considered good, when others would consider this low or weak agreement. How does this impact the results and the study limitations? A justification for the "bespoke checklist" is needed. The authors should discuss how this impacts the study especially in terms of design and rigor. Finally there remains some issues concerning the diagnostic accuracy levels used. The authors state that they have used an 80% diagnostic accuracy rating and have labeled this as a "good" accuracy level. However, work by Plante and Vance (1994) as well as other state that 80% accuracy is fair and 90% is good. I would suggest that the authors do a more complete reading and justification of this decision in their study design.

We agree with the reviewer that the kappa levels vary substantially, and in some places are fairly low, however, as the editor points out, this is not uncommon in this type of review, and we have been as rigorous as possible in addressing any disagreements. As stated above we have added some further detail on disagreement and coding procedures. We have also adjusted the terms used to more accurately reflect our cited source, although we appreciate others have differing thoughts on the labelling of the different levels. 

With regards to the diagnostic accuracy labels, we have changed this in the summary measures text to accurately reflect Plante and Vance, and we have checked in the results that this does not alter any results discussed.

Reviewer #2: Thank you for your additionnal work on this paper, which makes the message clearer.

however, "baby-tests" does non refer to newborn examination, but to developmental scales, developped from Gesell works. Baby-tests include Balyley scales for example.

I understood that you chose not to include them because they cannot be part of universal screening programs. Is it true? I may be worth specify.

Thank you for the clarification. These sorts of tests, such as the Bayley Scales, potentially could have been included as our criteria state that we were looking for papers involving universal assessment of children as part of whole population child health programmes or population based cohort studies. Within that, we were then broadly agnostic as to the method of development assessment used, but it is inevitably the case that the kinds of studies which were included in these types of studies will tend to involve relatively brief assessment methods that can be applied at scale, rather than more detailed assessments used for review of children at high risk of/with suspected developmental problems. The Bayley Scales offer a fairly detailed assessment of children’s developmental status, and we did not identify any studies meeting our wider inclusion criteria that used that specific approach to developmental assessment.

---

## [Decision Letter · Decision Letter 2]

5 Feb 2021

The predictive value of universal preschool developmental assessment in identifying children with later educational difficulties: a systematic review

PONE-D-20-12370R2

Dear Dr. Marryat,

We’re pleased to inform you that your manuscript has been judged scientifically suitable for publication and will be formally accepted for publication once it meets all outstanding technical requirements.  Although one reviewer requested additional clarifications about the steps involved in study selection and data extraction, neither the other reviewer nor I saw these as especially unclear or in need of improvement. 

Kind regards,

Matthew L. Hall, PhD

Academic Editor

PLOS ONE

Additional Editor Comments (optional):

Reviewers' comments:

Reviewer's Responses to Questions

**Comments to the Author**

1. If the authors have adequately addressed your comments raised in a previous round of review and you feel that this manuscript is now acceptable for publication, you may indicate that here to bypass the “Comments to the Author” section, enter your conflict of interest statement in the “Confidential to Editor” section, and submit your "Accept" recommendation.

Reviewer #1: All comments have been addressed

2. Is the manuscript technically sound, and do the data support the conclusions?

Reviewer #1: Yes

3. Has the statistical analysis been performed appropriately and rigorously? 

Reviewer #1: Yes

4. Have the authors made all data underlying the findings in their manuscript fully available?

Reviewer #1: Yes

5. Is the manuscript presented in an intelligible fashion and written in standard English?

Reviewer #1: Yes

6. Review Comments to the Author

Reviewer #1: Overall the authors have done a very good job responding to and revising the manuscript. Some minor concerns still exist in the clarity of the study selection section as well as measures of reliability for the data extraction section of the study. Specifically, it is difficult to follow the steps that were used to ensure the correct studies were selected. It is suggested that the authors think about writing this section as a series of steps so that this methodology can be replicated. Also, did the authors use any reliability measures to ensure the correct data was extracted from the study? This is currently unclear given how this section of the manuscript is written.

7. PLOS authors have the option to publish the peer review history of their article (what does this mean?). If published, this will include your full peer review and any attached files.

Reviewer #1: No

---

## [Editor Report · Acceptance letter]

12 Feb 2021

PONE-D-20-12370R2 

The predictive value of universal preschool developmental assessment in identifying children with later educational difficulties: a systematic review 

Dear Dr. Marryat:

I'm pleased to inform you that your manuscript has been deemed suitable for publication in PLOS ONE. Congratulations! Your manuscript is now with our production department. 

Kind regards, 

on behalf of

Dr. Matthew L. Hall 

Academic Editor

PLOS ONE